# A Compact Piezo-Inertia Actuator Utilizing the Double-Rocker Flexure Hinge Mechanism

**DOI:** 10.3390/mi14061117

**Published:** 2023-05-26

**Authors:** Pingping Sun, Chenglong Lei, Chuannan Ge, Yunjun Guo, Xingxing Zhu

**Affiliations:** 1School of Physics and Information Engineering, Jiangsu Second Normal University, Nanjing 211200, China; leichenglong@nju.edu.cn (C.L.); gechuannan@163.com (C.G.); guoyunjun09@aliyun.com (Y.G.); 2School of Electrical Engineering, Southeast University, Nanjing 210096, China

**Keywords:** piezo-inertia actuator, stick-slip, slip-slip, flexure hinge

## Abstract

With a simple structure and control method, the piezo-inertia actuator is a preferred embodiment in the field of microprecision industry. However, most of the previously reported actuators are unable to achieve a high speed, high resolution, and low deviation between positive and reverse velocities at the same time. To achieve a high speed, high resolution, and low deviation, in this paper we present a compact piezo-inertia actuator with a double rocker-type flexure hinge mechanism. The structure and operating principle are discussed in detail. To study the load capacity, voltage characteristics, and frequency characteristics of the actuator, we made a prototype and conducted a series of experiment. The results indicate good linearity in both positive and negative output displacements. The maximum positive and negative velocities are about 10.63 mm/s and 10.12 mm/s, respectively, and the corresponding speed deviation is 4.9%. The positive and negative positioning resolutions are 42.5 nm and 52.5 nm, respectively. In addition, the maximum output force is 220 g. These results show that the designed actuator has a minor speed deviation and good output characteristics.

## 1. Introduction

Precision positioning has been used in many microprecision applications, such as white light interferometry [1,2,3], atomic force microscopy [4,5], fiber alignment [6,7,8], and micro/nano equipment and manufacturing [9,10,11]. These applications usually require a micropositioner to achieve ultra-high resolutions and sufficient strokes at high speeds. The inherent advantages of the piezo-actuators allow fairly high speeds and nanoscale microsteps. They can be roughly divided into three categories, namely, ultrasonic actuators, inchworm actuators, and inertial actuators. An ultrasonic actuator is a kind of actuator developed based on the vibration of elastomers in the ultrasonic frequency band. The vibration of the elastomers is transformed into the continuous motion of the mover through friction [12,13]. This type of actuator can generally achieve higher speeds and displacements. An inchworm actuator is a kind of actuator that imitates the crawling action of multi-legged insects. This kind of actuator works at non-resonant frequencies, and features low speed, small size, great output force, and high displacement resolution [14,15]. Piezo-inertia actuators, also known as piezo-stick-slip actuators, utilize the moving part’s inertia to generate small displacements through uninterrupted frictional contact. The first piezo-stick-slip actuator was presented by Pohl in 1986 [16,17]. Subsequently, researchers from all over the world have developed many stick-slip actuators.

In practical applications, in particular the fast focusing imaging systems in white light interferometers, there is usually a requirement for a microlocator in order to ensure precise positioning. There are a number of requirements for the microlocator, such as low deviation between positive and reverse velocities, a compact structure, a high resolution, and a high speed. Piezo-inertia actuators can be seamlessly integrated into microprecision equipment to cater to motion needs in diverse scenarios. However, most existing piezoelectric inertial actuators cannot meet the above conditions at the same time. This is because the flexible hinge structure of the actuator cannot provide enough rebound force in the contraction stage of the piezo-stack to follow the movement of the piezo-stack, and cannot return to the original position at the same time, resulting in inconsistent positive and negative motion characteristics and low output [18]. Compared with the piezo-stack, the flexible hinge structure has low rigidity. By improving the stiffness of the flexible hinge structure, it is possible to achieve higher output characteristics at relatively high frequencies.

Many previous piezo-inertia actuators were developed to meet different requirements, such as Wang et al. [19], who designed a skateboard-type piezoelectric linear inertial actuator. The maximum forward and reverse motion speeds were 7.613 mm/s and 10.058 mm/s, respectively, with a deviation of 27.67% between positive and reverse speeds and forward and reverse displacement resolutions of 47 nm and 45 nm, respectively. However, the long strip flexure driving foot mechanism leads to excessive external dimensions. Li et al. [20] proposed a compact piezo-driven platform with an L-shape flexure hinge mechanism and maximum forward and reverse motion speeds of 1.648 mm/s and 1.403 mm/s, respectively, a deviation of 16.06% between positive and reverse velocities, and forward and reverse displacement resolutions of 17.9 μm and 15.3 μm, respectively. Shi et al. [21] designed a compact inertia piezoelectric actuator with a resolution of 4 nm, forward and reverse motion speeds of 2.5 mm/s and 1.8 mm/s, respectively, and a deviation of 32.56%. The speeds of the above two compact actuators are too low to come into practical use for fast focusing imaging systems in white light interferometers. It is worth noting that the piezo-stack nested in the flexure hinge mechanism is parallel to the moving platform, which provides a new idea for achieving a compact structure.

The purpose of this study is to achieve balanced output characteristics of a piezo-inertial actuator in terms of speed deviation, displacement resolution, and motion speed. Inspired by the movement mode of the double rocker mechanism, a double rocker-type flexure hinge mechanism actuator with high stiffness is proposed. In this paper, the structure design, analysis, and operation principle of the designed actuator are discussed in detail, and its output characteristics are tested. Forward and reverse motion with high consistency and good linearity are obtained. At last, the output characteristics of the actuator are compared with the performance of existing actuators, proving the advanced nature of the designed actuator and its potential to help improve practical engineering applications in white light interferometer systems.

## 2. Design and Analysis

At a fixed voltage, the piezo-stack can only extend in one direction, and the output displacement of the piezo-stack is typically limited to within a few tens of microns. To increase the output displacement of the piezo-stack, and thereby increase the velocity of the actuator in terms of the working stroke, previous researchers have proposed many working principles [22]. Among these, the stick-slip principle and the slip-slip principle have been widely employed to realize high resolution and high speed of the actuator, respectively. Hence, we chose these two principles as the driving modes of the actuator in this study.

The next task is to amplify the output displacement of a single piezo-stack and compact the actuator. A flexure hinge mechanism is usually required to amplify the output displacement of the piezo-stack and adjust the moving direction of the actuator. Inspired by the movement mode of the double rocker mechanism, we propose a high-rigidity flexure hinge mechanism similar to a double rocker mechanism in this study. The flexure hinge mechanism allows the piezo-stack to move in parallel to the moving platform thanks to ample free space in the interior of this type of flexure mechanism, providing a new way to achieve a compact actuator structure. At the same time, the flexible hinge mechanism with high stiffness can obtain high forward and reverse speeds and low speed deviations. Here, we choose the double rocker-type flexure hinge mechanism for the main structure of the actuator.

The double rocker mechanism is illustrated in Figure 1. The mechanism has two rockers, two connecting rods, and four pivot points. There is considerable free space inside the double rocker mechanism. When connecting rod B is fixed and rocker A rotates at a certain angle θ around pivot point 1, rocker B immediately rotates at the same angle around pivot point 4 after rocker A, and connecting rod A rotates and translates around pivot points 1 and 4 to location 2′-3′. According to observations of the trajectory of connecting rod A, pivot points 2 and 3 have the same horizontal displacement direction, while the vertical displacement direction is opposite.

### 2.1. Structure of Actuator

The construction and deformation of the double rocker-type flexure hinge actuator are illustrated in Figure 2. The overall dimensions of the actuator are 27.6 × 21.6 × 3 mm^3^. The actuator has one piezo-stack, two wedges, three rigid rods, five right circular flexure hinges, two linear-type flexure hinges, and one high-stiffness mounting part. All the parts except the piezo-stack, the wedges, and the friction head are part of a monolithic structure, which is the so-called “flexure hinge mechanism”. The right circular flexure hinges work as pivot points, and the rigid rods A and C are equivalent to the rockers. Rigid rod B and the mounting part work as the connecting rods. It is obvious from Figure 1 that when the friction contact is as close as possible to pivot point 2, this is conducive to improving the friction transmission efficiency. Therefore, the friction head is placed at the right end of rigid rod B of the flexure hinge mechanism and glued to the upper end of the flexure hinge mechanism. There is a right circular flexible hinge at the left end of rigid rod B. Its main purpose is to reduce the rotation stiffness between rigid rods A and B without affecting the axial stiffness of rigid rod B. This achieves an excellent value for the angle θ, providing the friction head with a great horizontal displacement $U_{x}$. Therefore, compared with the principle in Figure 1, the flexible hinge mechanism shown in Figure 2 has five pivot points (right circular flexible hinges). The piezo-stack (2 × 3 × 10 mm^3^, AE0203D08H09DF, NEC, Tokyo, Japan) nested inside the flexure hinge actuator is parallel to rigid rod B, which supplies power for the actuator. Two wedges are used to adjust the preload between the piezo-stack and the double rocker-type flexure hinge mechanism. The piezo-stack has a first end bonded to rigid rod C and a second end bonded to the mounting part. The two linear-type flexure hinges attached to the two sides of the piezo-stack are used to reduce the shear stress applied to the piezo-stack in order to protect it.

Figure 3 shows the detailed construction of the moving platform driven by the designed actuator, which consists of a double rocker-type flexure hinge actuator, a moving platform, a base, a preload screw, and two mounting screws. The normal force between the moving platform and the friction head can be adjusted by the preload screw. Therefore, the moving platform is capable of self-locking through the static friction between the friction head and the moving platform. When the normal force is adjusted to an appropriate level, the designed actuator is fixed to the base by two mounting screws.

Assuming that point A in Figure 1 is the output displacement action point of the piezo-stack to rigid rod C, $l_{12}$ is the distance between the center of the right circular flexure hinges at both ends of rigid rod C, and $l_{1A}$ is the distance between point A and the center of pivot point 1 (right circular flexure hinge), when the piezo-stack operates at 100 V, it extends over its length and causes the flexure hinge mechanism to deform in both the x and y directions. The displacement in the y direction changes the normal force (20 N) between the friction head and the moving platform. The normal force changes with the rotation angle of rigid rod C. The displacement $l_{x}$ generated by the piezo-stack is the motion displacement, the displacements $U_{x}$ and $U_{y}$ are the motion displacements of the friction head, and θ is the rotational angle of rocker A (rigid rod C). All of the above parameters satisfy the following equations:(1)$$\frac{l_{x}}{U_{x}}=\frac{l_{1A}}{l_{12}}$$
(2)$$U_{x}=l_{12}\sin\theta$$
(3)$$U_{y}=l_{12}(1-\cos\theta )$$
(4)$$\Delta F=Kl_{x}\sin\theta$$

In this study, the values of $l_{x}$, $l_{1A}$, and $l_{12}$ are 6.1 μm, 5.5 mm, and 10.3 mm, respectively, ΔF denotes the fluctuation of the normal force, and the stiffness *K* of the piezo-stack is 22 N/μm. According to the Equations (1), (2) and (4), the fluctuation of the normal force is 0.149 N, which is far smaller than the normal force 20 N. Therefore, the actuator can maintain stable operation.

### 2.2. Operating Principle of Actuator

According to the movement mode of the double rocker mechanism and the structure of the double rocker-type flexure hinge actuator, the motion of the designed compact piezo-inertia actuator in a single working circle corresponding to a saw-tooth driving voltage is shown in Figure 4 and Figure 5. A saw-tooth driving voltage contains two phases: the linear slow rise phase t_0_ − t_1_ and the sharp drop phase t_1_ − t_2_, or the sharp rise phase t_0_ − t_1_ and the linear slow drop phase t_1_ − t_2_. Based on the forward and reverse motion processes of the designed compact piezo-inertia actuator, as shown in Figure 4 and Figure 5, the motion processes can be divided into three steps.

Step 1: the piezo-stack and the moving part are at the home position, as the piezo-stack is not powered.

Step 2: In Figure 4b, under a slow-rising linear voltage, the piezo-stack pushes the flexure hinge mechanism, causing the friction head to stick with the moving platform and rub the moving platform to make it move forward by static friction force. The moving platform moves forward by a step $\Delta S_{F}$. This is the so-called “stick” process.

Step 3: In Figure 4c, the voltage sharply drops to zero, the piezo-stack and the friction head abruptly contract to its home position, and the friction head slips rapidly backward relative to the moving platform and rubs the moving platform to make it move backward through the sliding friction force. The moving part moves backward by a small step $\Delta S_{B}$. This is the so-called “slip” process. Due to the inertia of the moving platform, the backward step of the moving platform during the slip process is smaller than the forward step during the stick process.

In this way, a “stick-slip” movement cycle of the piezo-inertia actuator ends up with a forward step. This is the so-called “stick-slip” process. The forward step in a full cycle can be calculated using the following equation:(5)$$\Delta L_{F}=\Delta S_{F}-\Delta S_{B}$$

For reverse motion, the motion process turns inversely to the “slip-stick” mode as shown in Figure 5a,b. The reverse motion step in a whole cycle can be calculated using the following equation:(6)$$\Delta L_{B}=\Delta S_{B}-\Delta S_{F}$$

To achieve high-resolutions (known as the minimum displacement step), the “stick-slip” and “slip-stick” principle can be used to achieve small positive and negative displacement steps of the actuator. By repeating an entire cycle, a continuous stepping motion can be realized in both the positive and negative directions.

When driving the actuator at a high frequency, the “stick” process disappears completely in the forward and reverse motion, and the whole driving principle turns into the “slip-slip” mode. By repeating the “slip-slip” cycle, it is possible to achieve a good stroke at high speed in the positive or negative direction.

### 2.3. Simulation of Flexure Hinge Mechanism

The designed flexure hinge actuator was simulated using the finite element method (FEM). AL7075, which has excellent elasticity, was chosen as the material for the flexure hinge mechanism. The elastic modulus, Poisson’s ratio, and density of AL7075 are 72 GPa, 0.33, and 2810 kg m^−3^, respectively. The friction head, which is abrasion resistant, is a key component in friction conversion. The smooth dry hard surface of ZrO_2_ is a suitable material for friction heads. The elastic modulus, Poisson’s ratio, and density of ZrO_2_ are 320 GPa, 0.25, and 6020 kg m^−3^, respectively. The mounting part of the flexure hinge mechanism was completely secured during simulation. The material constants of the piezo-stack are shown in Table 1. As the excitation voltage was 100 V DC, the piezo-stack extends by a stroke of approximately 6.1 μm, resulting in the simulated displacement of the friction head in the x and y positive directions being about 10.73 μm and 1.65 μm, respectively, as shown in Figure 6. The numerical solution displacements of the friction head in the x and y positive directions can be calculated through Equations (1)–(3), for which the results are about 11.42 μm and 0.0063 μm, respectively. It can be noted that the displacement of the simulation value in the x direction is close to the numerical solution, while the displacement error in the y direction is greater. This is mostly because a rigid rod with two right circular flexure hinges is equivalent to a rigid body in the numerical solution. In this study, the displacement in the x direction of the flexure hinge mechanism is used to drive the moving platform to move along the x direction; hence, the y direction displacement can be ignored.

During the operating process of the actuator, its operating frequency should be kept out of the resonance frequency range to ensure that no abnormalities occur as a result of the resonance of the actuator. Figure 7 shows the result of the modal analysis in the range of 0 to 5000 Hz. There are two resonance frequency points, 1660.8 Hz and 3146.7 Hz. Therefore, the working frequency selected should be much lower than 1660.8 Hz.

## 3. Experiments and Results

### 3.1. Experimental System

Figure 8a,b shows the experimental setup used to measure the mechanical properties of the designed piezo-inertia actuator. The experimental setup consisted of a signal generator, an oscilloscope, an amplifier, a laser sensor, and a controller. The required saw-tooth voltage signal was generated by a signal generator (SDG1005; SIGLENT TECHNOLOGIES Co., Ltd., Shenzhen, China) and further amplified by a power amplifier (E05.A3, Harbin Core Tomorrow Science & Technology Co., Ltd., Harbin, China). The required voltage value and frequency applied to the actuator were monitored by an oscilloscope (TDS2024; TEKTRONIX Inc., Beaverton, OR, USA). The output displacement of the moving platform was measured by the combination of a laser sensor and controller (LK-H020/LK-H150/LK-G5001P; Keyence Co., Osaka, Japan), and all measured data were collected by a computer.

The actuator was successfully assembled, as shown in Figure 8c. We measured the static friction between the moving platform and the friction head by pulling the moving platform through the mechanical tension meter. Using Equation (7), the normal force *F* can be calculated, where *f* and *μ* denote the static friction and the static friction coefficient, respectively. The measured static friction force was 3 N and the static friction coefficient was 0.15. Therefore, the normal force was calculated as 20 N.
(7)f=μ⋅F

### 3.2. Output Characteristics under Various Driving Frequencies

The output characteristics at a driving frequency between 1–1500 Hz were measured for the positive and negative directions. A saw-tooth driving voltage with degree of symmetry 98%, illustrated in Figure 4, was fixed at 100 V, and was then used to drive the forward motion of the actuator in the subsequent experiments. For reverse motion, the degree of symmetry of the driving voltage was set as 2%.

Figure 9 shows the positive and negative output displacement characteristics within four seconds at a driving frequency of 1 Hz. The forward and backward motion at each step can be easily observed, and there is a stepwise increase or decrease. The output displacement of every step in the positive and negative directions can be calculated with Equations (4) and (5), respectively; the cumulative positive and negative output displacements within 4 s were 0.17 μm and 0.21 μm. Therefore, the average output step displacements in the positive and negative direction were 42.5 nm and 52.5 nm, respectively.

Figure 10 shows the positive and negative output displacement characteristics at a driving frequency between 10–1500 Hz. When the driving frequency increases from 10 to 1000 Hz, the cumulative output displacement increases from 19.84 μm to 10.63 mm for the positive direction and from 18.74 μm to 10.12 mm for the negative direction. As the frequency exceeds 1000 Hz, the positive and negative output displacement gradually decreases. This is because the frequency band is close to the first resonance frequency range of the actuator, as shown in Figure 7.

The stepwise motion in both directions only occurs below the driving frequency of 30 Hz, which is in the “stick-slip” or “slip-stick” process, as shown in Figure 10a. When the driving frequency is over 40 Hz, the movement of the ladder shape is no longer obvious, and the movement turns into a sliding mode (“slip-slip” mode). Therefore, the motion processes in both directions are consistent with the analysis in Figure 4 and Figure 5.

Figure 11 shows the relationship between the positive and negative velocities of the actuator and the driving frequency, respectively. As the driving frequency increases, the velocity of the actuator increases. The velocity of the actuator along the positive direction is 10.63 mm/s under a driving frequency of 1000 Hz, and is 0.0425 μm/s under 1 Hz. In the negative direction, the corresponding velocities are 10.12 mm/s and 0.0525 μm/s.

All the output displacement curves have good linearity at a driving frequency between 10–1500 Hz, demonstrating that the actuator has excellent operating stability in open-loop mode. However, the open-loop output characteristics in the positive and negative directions are slightly different. This can be easily solved with a closed-loop mode. The maximum positive and negative velocities are 10.63 mm/s and 10.12 mm/s, respectively, and the respective minimum positive and negative stepping displacements (known as the resolution) are about 42.5 nm and 52.5 nm.

### 3.3. Output Characteristics at Various Driving Voltages

Section 3.2 covers the relationship between output characteristics and driving frequency. In this section, the influence of the driving voltage on output characteristics is discussed. The stepwise motion obtained at 30 Hz is easier to observe and compare. When the driving frequency is higher than 30 Hz, the stepwise motion is less evident. Therefore, we chose 30 Hz and 40 Hz as the driving frequencies in the following experiment, and increased the driving voltage from 30 to 100 V in increments of 10 V.

Figure 12a shows the positive and negative output displacement characteristics when the driving voltage ranges between 50–100 V at a driving frequency of 30 Hz. The stepwise motion shows very good linearity at various driving voltages. When the driving voltage increases from 50 V to 100 V, the positive cumulative output displacement increases from 7.42 μm to 64.69 μm, and the negative cumulative output displacement increases from 7.23 μm to 74.62 μm. As the driving voltage is less than 50 V, the stepwise motion becomes very unstable, which is not suitable for practical engineering applications.

Figure 12b shows the positive and negative output displacement characteristics when the driving voltage increases from 30 V to 100 V at a driving frequency of 40 Hz. The positive cumulative output displacement increases from 23.21 μm to 209.85 μm, and the negative cumulative output displacement increases from 23.32 μm to 184.31 μm. Figure 13 shows the relationship between the positive and negative velocity of the actuator and the driving voltage. At a driving frequency between 30 Hz and 40 Hz, the speed of the actuator increases with the increase in the driving voltage.

From the above experimental results, it can be seen that the driving voltage is an important factor that affects the output characteristics of the actuator.

### 3.4. Load Characteristics

Load capacity is one of the most important parameters in the design of an actuator. Therefore, the impact of the load capacity on output characteristics is further discussed in this section. Figure 14a shows the test system for the load capacity of the actuator. When the driving frequency is fixed at 1000 Hz and the driving voltage is fixed at 100 V, as the external load increases from 0 to 220 g, the positive speed decreases from 10.63 mm/s to 1.12 mm/s and the negative speed decreases from 10.12 mm/s to 1.05 mm/s. The maximum external load of the actuator in both the positive and negative directions is about 220 g. For the application of fast focusing imaging systems in white light interferometry, a lens with a weight of 50 g is used as an external load, with corresponding positive and negative speeds of 9.15 mm/s and 9.05 mm/s, respectively. This arrangement can meet the needs of highly dynamic applications.

## 4. Comparison and Discussion

Table 2 compares previously developed actuators with the one developed in this work in terms of driving voltage, driving frequency, number of driving voltages, maximum forward and reverse speed, forward and reverse speed deviation, displacement resolution, forward and reverse output load, and overall size of the actuator. In all the above-mentioned inertial actuators, one or two piezoelectric stacks are embedded in the flexible hinge structure by mechanical preloading. The piezo-stacks are driven by saw-tooth waveform voltages. In [23], Koc et al. proposed a two-phase inertial piezoelectric motor. Through two-phase voltage driving in the ultrasonic frequency band, the hysteresis nonlinearity of each of the two piezoelectric stacks is offset by the other, ensuring that the forward and reverse speeds stay consistent and high. Other previous actuators, as well as the one developed in the present study, operate in non-ultrasound frequency ranges (i.e., the operating frequency does not exceed 3000 Hz), and are driven by a single-phase voltage. In comparison to a single-phase drive actuator, a two-phase drive actuator has a lower driving voltage, higher speed, and greater load capacity (the load direction being the same as the speed direction). However, considering the design of two-phase drive actuators and the ultrasonic frequency range, the design of the drive circuit is more complex and the size must be increased.

Compared with all the single-phase drive actuators, the actuator designed in the present study has significantly improved positive and reverse speeds and good consistency. This is sufficient to show that the designed double rocker flexible hinge mechanism has high structural stiffness. The maximum output load of the actuator is 2.2 N (220 g), which is slightly better than that of the other actuators. The positive and reverse displacement resolutions and structure size of the actuator are competitive compared to similar actuators.

Through the above comparison, although the developed actuator has improved performance, the speed is lower than that of the compared two-phase drive actuator. Therefore, in future research the speed deviation needs to be further reduced through connection of the actuator in parallel, as shown in Figure 15. In the image, the actuator is driven by two saw-tooth waveform voltages. In this way, the speed and load can be doubled. Because it operates in the non-resonant frequency band, the driving circuit is simplified.

## 5. Conclusions

In summary, a compact piezo-inertia actuator was designed based on the movement of a double rocker mechanism. Compared with previous actuators, the designed actuator has significantly improved forward and reverse maximum motion speeds and better deviation between the positive and negative speeds, demonstrating the advanced nature of this design. The experimental results show that both forward and reverse motions have high consistency and good linearity at various driving frequencies and voltages, indicating that the designed actuator has good motion stability. At a voltage of 100 V, the maximum positive and negative velocities are about 10.63 mm/s and 10.12 mm/s, respectively, and the corresponding speed deviation is 4.9%. The positive and negative displacement resolutions are about 42.5 nm and 52.5 nm, respectively. In addition, the maximum output force is 2.2 N (220 g). This study confirms that the design with a compact double rocker-type flexure hinge mechanism can provide new ideas for the further development of piezo-inertia actuators with high resolution, high speed, and low deviation. In addition, we intend to apply the proposed design to the fast focusing imaging system of white light interferometers.

## Figures and Tables

**Figure 1 micromachines-14-01117-f001:**
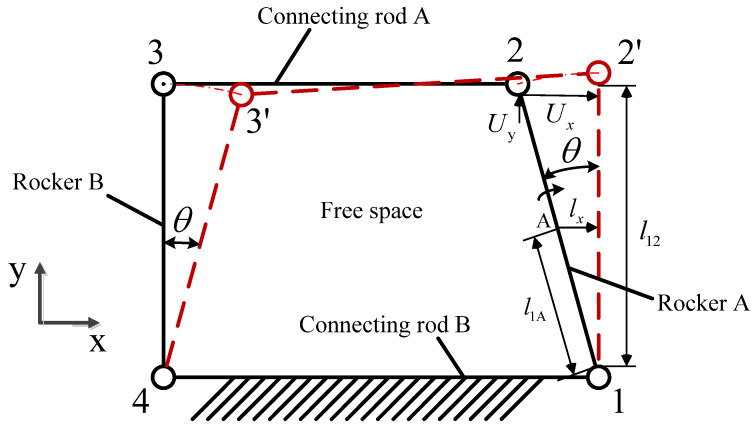
Schematic diagram of double rocker mechanism.

**Figure 2 micromachines-14-01117-f002:**
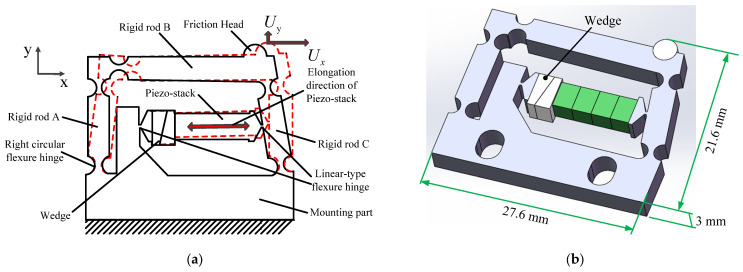
Structure of actuator: (**a**) schematic diagram and (**b**) three-dimensional model of actuator.

**Figure 3 micromachines-14-01117-f003:**
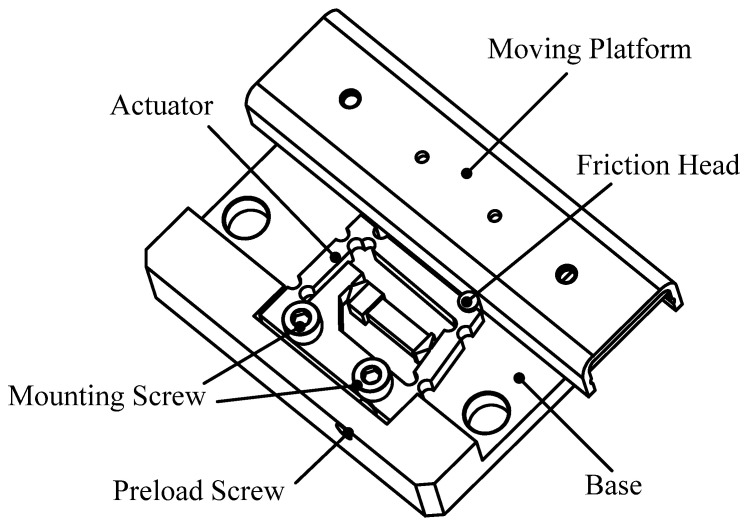
Construction of the moving platform driven by the designed actuator.

**Figure 4 micromachines-14-01117-f004:**
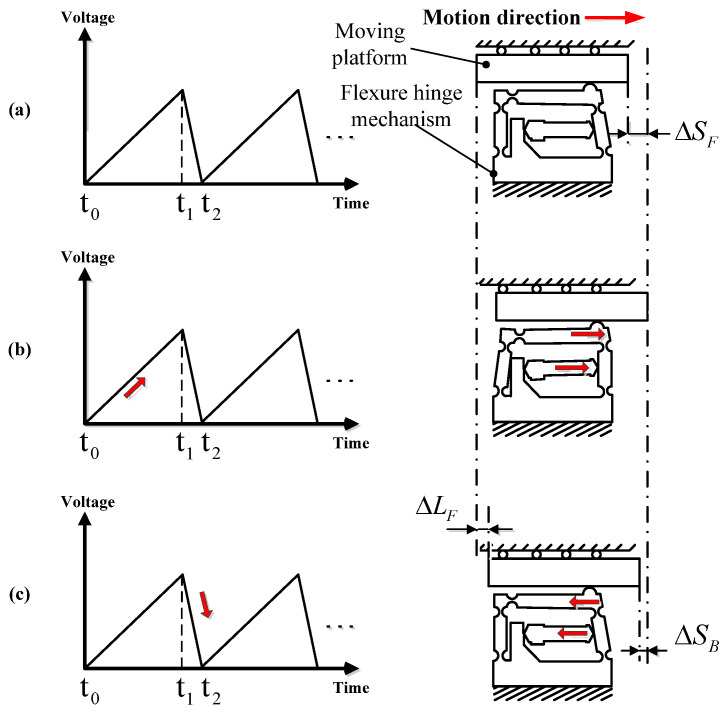
Forward motion: (**a**) Step 1; (**b**) Step 2; (**c**) Step 3.

**Figure 5 micromachines-14-01117-f005:**
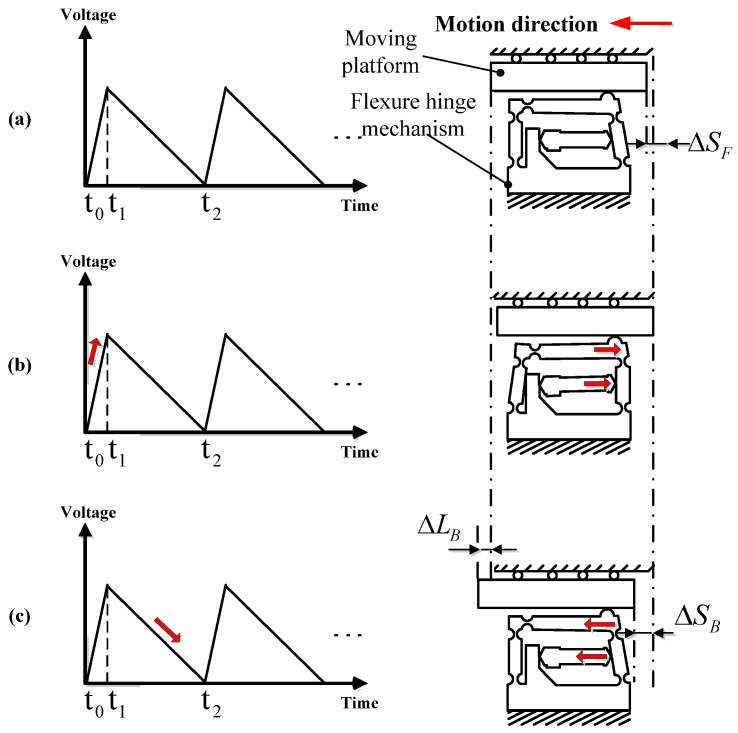
Reverse motion: (**a**) Step 1; (**b**) Step 2; (**c**) Step 3.

**Figure 6 micromachines-14-01117-f006:**
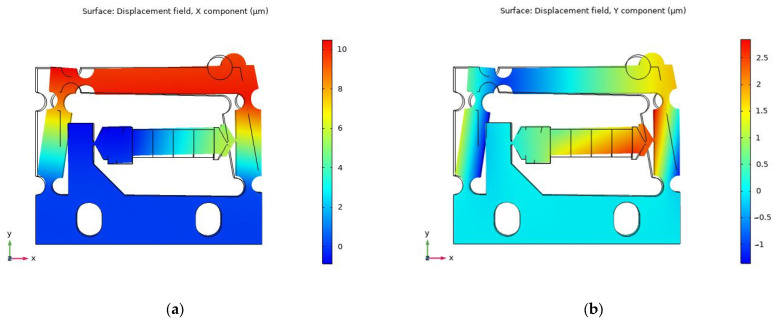
Simulation of the designed flexure hinge actuator: (**a**) X-direction displacement and (**b**) Y-direction displacement.

**Figure 7 micromachines-14-01117-f007:**
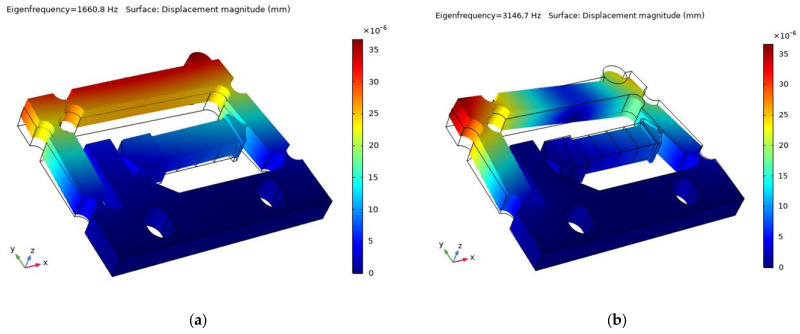
Modal analysis: (**a**) 1660.8 Hz and (**b**) 3146.7 Hz.

**Figure 8 micromachines-14-01117-f008:**
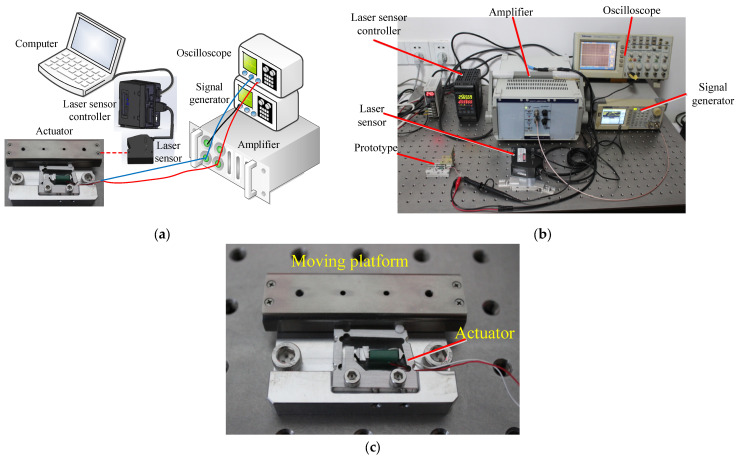
Experimental system of the designed actuator: (**a**) schematic diagram of experimental system; (**b**) experimental system; (**c**) moving platform driven by the designed actuator.

**Figure 9 micromachines-14-01117-f009:**
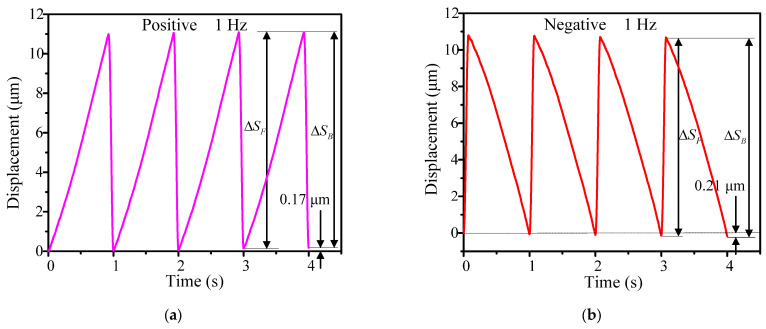
Output displacements in four steps under the driving frequency of 1 Hz: (**a**) positive direction and (**b**) negative direction.

**Figure 10 micromachines-14-01117-f010:**
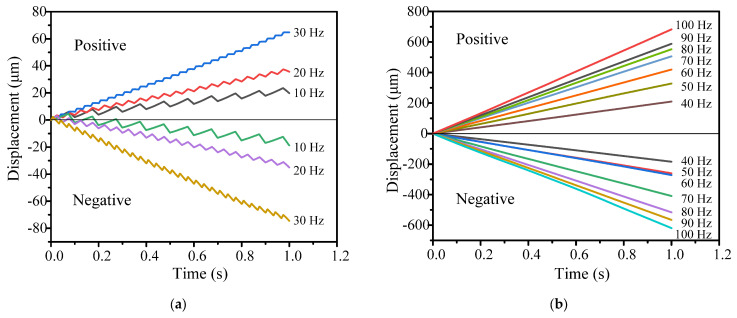
Output displacement performance under various frequencies: (**a**) 10–30 Hz and (**b**) 40–100 Hz; (**c**) positive displacement in the range of 200 Hz to 1500 Hz; (**d**) negative displacement in the range of 200 Hz to 1500 Hz.

**Figure 11 micromachines-14-01117-f011:**
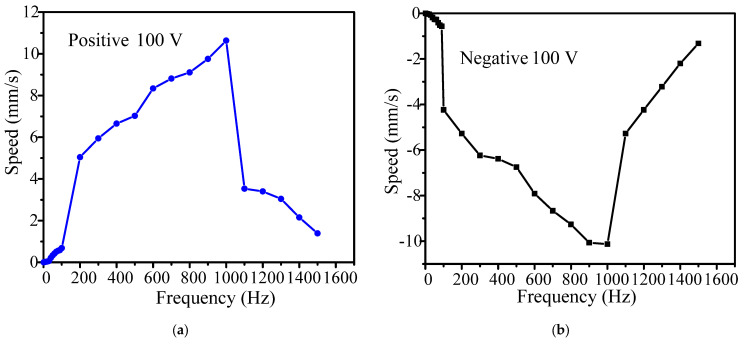
Relationship between speed and driving frequency: (**a**) positive direction and (**b**) negative direction.

**Figure 12 micromachines-14-01117-f012:**
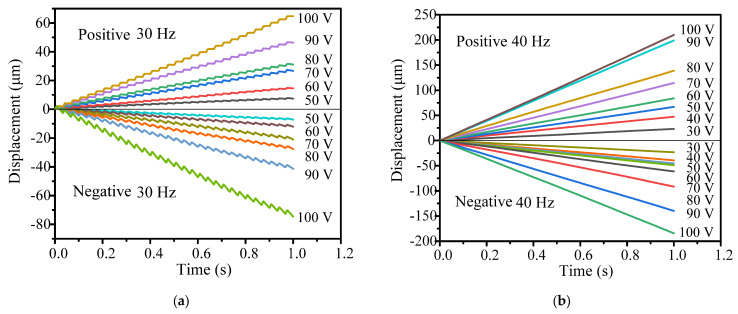
Output displacements under various driving voltages: (**a**) 30 Hz and (**b**) 40 Hz.

**Figure 13 micromachines-14-01117-f013:**
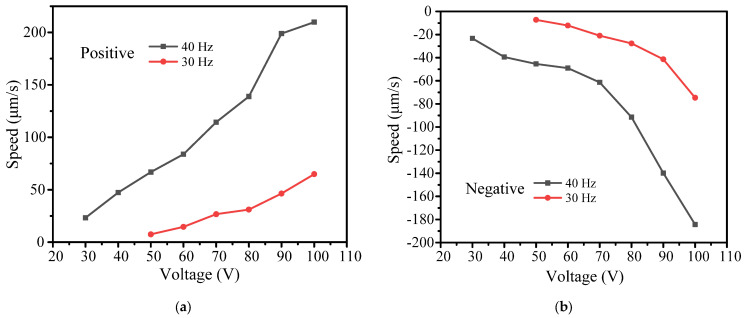
Relationship between speed and driving voltage: (**a**) positive direction and (**b**) negative direction.

**Figure 14 micromachines-14-01117-f014:**
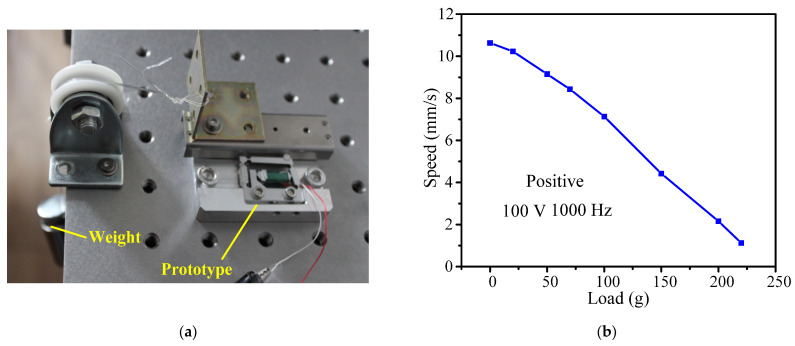
Loading characteristic: (**a**) experimental system; (**b**) positive direction; (**c**) negative direction.

**Figure 15 micromachines-14-01117-f015:**
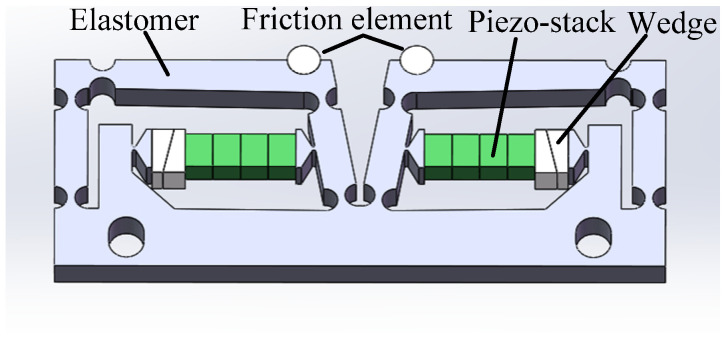
Parallel-type actuator.

**Table 1 micromachines-14-01117-t001:** Material constants of the piezo-stack.

Material	PZT-5H
Density (kg/m^3^)	7500
Poisson’s ratio	0.3
Elastic modulus (×10^10^ N/m^2^)	$\left(\begin{array}{cccccc} 12.72 & 8.02 & 8.47 & 0 & 0 & 0\\ 8.02 & 12.72 & 8.47 & 0 & 0 & 0\\ 8.47 & 8.47 & 11.74 & 0 & 0 & 0\\ 0 & 0 & 0 & 2.29 & 0 & 0\\ 0 & 0 & 0 & 0 & 2.29 & 0\\ 0 & 0 & 0 & 0 & 0 & 2.35 \end{array}\right)$
Piezoelectric constant (C/m^2^)	$\left(\begin{array}{ccc} 0 & 0 & -2.74\\ 0 & 0 & -2.74\\ 0 & 0 & 5.93\\ 0 & 7.41 & 0\\ 7.41 & 0 & 0\\ 0 & 0 & 0 \end{array}\right)$
Dielectric constant	$\left(\begin{array}{ccc} 1704.4 & 0 & 0\\ 0 & 1704.4 & 0\\ 0 & 0 & 1433.6 \end{array}\right)$

**Table 2 micromachines-14-01117-t002:** Comparison between previous actuators and the one developed in this work.

Literature	[19]	[20]	[21]	[23]	This Work
Driving voltage (V)	100	100	60	±25	100
Driving frequency (Hz)	280	100	2200	20,000	1000
Number of driving voltages	1	1	1	2	1
Maximum forward speed (mm/s)	7.613	1.648	2.5	16	10.63
Maximum reverse speed (mm/s)	10.058	1.403	1.8	/	10.12
Speed deviation	27.67%	16.06%	32.56%	/	4.9%
Forward displacement resolution (nm)	47	17,900	4	800	42.5
Reverse displacement resolution (nm)	45	15,300	/	800	52.5
Load (N)	2	2	/	3.2	2.2
Overall size of the actuator (mm^3^)	62.3 × 32 × 38	55 × 55 × 29.75	17 × 5 × 6	/	27.6 × 21.6 × 3

## Data Availability

Data underlying the results presented in this paper are not publicly available at this time, but may be obtained from the authors upon reasonable request.

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
