# Peer review of "A Compact Piezo-Inertia Actuator Utilizing the Double-Rocker Flexure Hinge Mechanism"

_micromachines, 2023, doi:10.3390/mi14061117_

Round 1

Reviewer 1 Report

In order to achieve high speed, high resolution, and low deviation, a dual rocker flexible hinge piezoelectric inertial actuator is proposed in the paper. The load capacity, voltage characteristics, and frequency characteristics of the actuator are studied, and a prototype is made for relevant testing. The paper has certain theoretical significance. Suggest receiving after modification.

Suggest improving or perfecting the paper in the following areas:

 1) How to solve the fatigue life problem of the linear hinges at both ends of the piezoelectric stack during frequent operation.

2) The actuator designed in the paper is in contact with the sliding platform during operation. When a certain frequency and amplitude of working voltage is applied to the piezoelectric stack, its expansion deformation drives the movement of the flexible hinge mechanism and achieves displacement amplification. The movement of the flexible hinge mechanism in the x-direction is driven by the friction head to move the sliding platform. The deformation in the y-direction of the flexible hinge mechanism causes the friction head to come into contact with the sliding platform, and the displacement in the y-direction follows θ The change in the friction head causes a change in the clamping force between the friction head and the sliding platform. If the clamping force is too small, the friction force between the friction head and the sliding platform may not be sufficient to drive the sliding platform to move. If the clamping force is too large, it will prevent the movement of the sliding platform or even get stuck. Please analyze the compression force as follows θ Analyze the laws of change and how to address the impact of clamping force changes on motion.

 3) When studying the influence of operating frequency characteristics on output displacement in the paper, it was found that when the operating frequency increases between 200Hz and 1000Hz, the output displacement increases with the increase of operating frequency. When the operating frequency is between 1000Hz and 1500Hz, the output displacement decreases with the increase of operating frequency. Please analyze the reasons for this change pattern.

Author Response

Thank you for giving us the chance to make our manuscript better. We are grateful to the reviewers for giving such valuable comments and suggestions. We have revised the manuscript according to the comments carefully and we hope the manuscript will meet the requirement of the magazine. Revised parts of the manuscript are marked in red. Our point-by-point response to the comments is as follows.Please find the attached file.

Author Response

(The authors gave the same response as above.)

Reviewer 3 Report

Dear Authors,

You could find my comments about your paper in the attachment.

Kind Regards

English should be polished

Author Response

Thank you for giving us the chance to make our manuscript better. We are grateful to the reviewers for giving such valuable comments and suggestions. We have revised the manuscript according to the comments carefully and we hope the manuscript will meet the requirement of the magazine. Revised parts of the manuscript are marked in red. At the same time, we have improved the writing of the paper. At the same time, we have improved the writing of the paper. Our point-by-point response to the comments is as follows. Please find the attached file.

Reviewer 4 Report

1. Chapter “Design and analysis” must be improved:
a) in line 115 the authors present the dimensions of the actuator (27.6x21.6x3 mm3). However, it is not known exactly what these dimensions refer to. These dimensions should be marked in Fig. 2 and 3,
b) the piezoelectric stack should be described in more detail:
- type of piezoelectric material,
- material properties of this piezoelectric material: d33, s33 and ε33 should be given,
c) the method of connecting the piezoelectric stack with the mechanical elements of the actuator should be more described in detail,
d) a diagram of connecting the piezoelectric stack with the mechanical elements should be added.

2. Chapter “Experiments and results” must be improved:
a) the methodology of experimental research should be more precisely described:
- a diagram of the laboratory stand should be provided. this diagram should show what exactly the laser sensor measures.

3. Chapter “Comparison and Discussion” must be improved:
a) In table 1, the authors present a comparison of selected aspects of the operation of actuators known from the literature versus actuator presented in this article. This comparison should be significantly supplemented with:
- the waveforms and values of input voltages to piezoelectric stack,
- an explanation of what the dimensions refer to,
- a description of the load acting on the actuator,
- the degree of complexity of the structure,
- method of connecting piezoelectric stacks with mechanical parts of actuators.
These aspects can be the basis for discussion, which is missing in this chapter. The authors must discuss the results, because they try to compare very different designs of actuators.

4. Chapter “Conclusions” must be improved:
a) conclusions must take into account the discussion of the results referred to in point 3 of this review.

Moderate editing of English language is needed.

Author Response

(The authors gave the same response as above.)

Round 2

Reviewer 3 Report

Dear Authors,

I do not have any other comments about the revised version of the paper.

Kind Regards

Reviewer 4 Report

I accept in present form.

Minor editing of English language required.